# An insight into the stochastic solitonic features of the Maccari model using the solver technique

Hesham G. Abdelwahed[1,2]*, Reem Alotaibi[1], Emad K. El-Shewy[2,3], Mahmoud A. E. Abdelrahman[4,5]

**1** Department of Physics, College of Science and Humanities, Al-Kharj, Prince Sattam bin Abdulaziz University, Al- Kharj, Saudi Arabia, **2** Theoretical Physics Group, Faculty of Science, Mansoura University, Mansoura, Egypt, **3** Department of Physics, College of Science, Taibah University, Al-Madinah, Al-Munawarah, Saudi Arabia, **4** Department of Mathematics, College of Science, Taibah University, Al-Madinah, Al-Munawarah, Saudi Arabia, **5** Department of Mathematics, Faculty of Science, Mansoura University, Mansoura, Egypt

* h.abdelwahed@psau.edu.sa, hgomaa_eg@mans.edu.eg

**Data Availability Statement:** All relevant data are within the manuscript.

**Funding:** The author(s) received no specific funding for this work.

## Abstract

In this paper, the unified approach is used in acquiring some new results to the coupled Maccari system (MS) in Itô sense with multiplicative noise. The MS is a nonlinear model used in hydrodynamics, plasma physics, and nonlinear optics to represent isolated waves in a restricted region. We provide new results with complicated structures to this model, including hyperbolic, trigonometric and rational function solutions. We draw the two dimensional (2D) and three dimensional (3D) graphs to some of the study's solutions under appropriately chosen physical parameter values. Random factors can alter the collapse caused by turbulence in the model medium. We noticed that our results may be useful for solving some real-world physical issues by identifying the motion of an isolated wave in a small area.

## 1 Introduction

Nonlinear partial differential equations (NPDEs) are widely used in the applied sciences to describe many complicated processes [1–6]. One of the areas that scientists find most intriguing in the current period is nonlinear phenomena. Different types of exact solutions to NPDEs, like soliton, negaton, peakon, explosive, cuspon, rational and periodic solutions, have attracted a lot of attention in recent years. These solutions play an essential role in the research of nonlinear physical processes. In the modern scientific and technical era, various researches have been employed to create several analytical processes to get solitary wave solutions for NPDEs [7–13].

A deterministic model consists of equations that explain the system's evolution throughout time. A random variable results from a stochastic process that reflects an observation at a certain point in time. In a stochastic model, evolution is somewhat random, and repeating the process may not provide the same outcomes. Deterministic models and stochastic models can be compared. The growth of chaotic models in recent years has somewhat blurred the line

**Competing interests:** The authors have declared that no competing interests exist.

between deterministic and stochastic models. A common stochastic process that combines the characteristics of a Markov process with a martingale is the Wiener process [14]. A popular stochastic process in dispersive situations is this one [15, 16]. Recent advances in stochastic calculus, especially in the context of stochastic partial differential equations (SPDEs), seem to us to provide the foundation for a comprehensive modeling of practical applications [15]. Mathematicians are the most comfortable using SPDEs and stochastic processes on natural models.

The dynamics of an isolated wave in a limited area of space is often described by the Maccari's system (MS), a kind of NPDEs utilized in many different domains, including materials science, acoustic waves in crystals, superfluid, nonlinear optics, and so forth [17–22]. Maccari developed this system from the Kadomtsev-Petviashvili equation by employing a reduction approach based on spatiotemporal rescaling [23]. He demonstrated how the MS accurately characterised the extremely significant aspects of rogue waves and how they might be utilized to examine diverse nonlinear forms such as standing waves, nonlinear optical fibres, and fluid dynamics [24]. The MS system might be applied to complicated systems to explore the dynamics of water and energy waves. Various mathematical approaches may be used to identify solitons and dark solitons, however rogue wave solutions have just been identified in the MS system [17–22]. These prior studies were all carried out from a deterministic perspective.

The coupled MS reads [18, 25, 26]:

$$i\,U_t + U_{xx} + \Phi U = 0\,,$$

$$\Phi_t + \Phi_y + \left(|U|^2\right)_x = 0,$$

(1.1)

$U = U(x, y, t)$, $\Phi = \Phi(x, y, t)$ denote the complex scalar field and real scalar field, respectively. Zhao [27] present various solitary wave solutions for model (1.1). Furthermore, many periodic and solitons of the above model have recently been found in [18, 25, 26, 28]. In this paper, we consider model (1.1) via Wiener process as follows [29]:

$$i\,U_t + U_{xx} + \Phi U - i\,\sigma\,U\,W_t = 0\,,$$

$$\Phi_t + \Phi_y + \left(|U|^2\right)_x = 0.$$

(1.2)

The noise $W_t$ is a Wiener times derivative of $W(t)$ and $\sigma$ denotes noise strength [30]. Specifically, we use a unified method to investigate this system via multiplicative noises in the Itô sense. This methodology provides a number of benefits over the bulk of existing methods, including the avoidance of difficult and calculations that take a long time and the generation of precise results. It is not difficult, dependable, and effective. This technique provides several sorts of solitary waves dependent on the physical parameters. The presented solutions have significant applications in hydrodynamics, optical fiber communications, and plasma physics. This approach may be used as a box-solver for several systems in natural science. To our knowledge, the proposed approach for solving the MS has never been applied previously.

A Wiener process is a stochastic process that is continuous across time. The primary characteristics of Brownian motion $\{W(t)\}_{t \geq 0}$ are shown as follows:

(a) $W(t)$, $t \geq 0$ are continuous functions of $t$ and $W(t) \sim N(0, t)$ for time $t$.

(b) For $0 \leq t_1 < t_2 < t_3 \ldots < t_n$; $W(t_2) - W(t_1)$; $W(t_3) - W(t_2)$;... $W(t_n) - W(t_{n-1})$ are independent.

(c) $W(t) - W(s)$ follows a normal distribution with zero mean and variance $t - s$, i.e.
$\Psi(t) - \Psi(s) \sim \sqrt{t - s}\,N(0, 1)$, where $N(0, 1)$ is a standard normal distribution.

The rest of the paper is constructed as follows. Section 2 briefly describes the unified technique. In Section 3, we introduce new stochastic solutions in the Itô sense for MS system. We demonstrate the physical behavior of the solutions and the influence of the noise factor on their evolution. Furthermore, graphs of specific achieved solutions are provided. Section 5 presents findings, perspectives, and recommendations for further work.

## 2 Unified solver method

We introduce a condensed version of the unified solver approach for the following equation [31]:

$$\Lambda_1 u'' + \Lambda_2 u^3 + \Lambda_3 u = 0. \tag{2.1}$$

The solutions of this equation are
(i) Hyperbolic solutions: (at $\frac{\Lambda_3}{\Lambda_1} > 0$)

$$u_{1,2}(x, t) = \pm \sqrt{\frac{-\Lambda_3}{\Lambda_2}} tanh\left(\sqrt{\frac{\Lambda_3}{2\Lambda_1}}(\zeta + \vartheta)\right) \tag{2.2}$$

and

$$u_{3,4}(x, t) = \pm \sqrt{\frac{-\Lambda_3}{\Lambda_2}} coth\left(\sqrt{\frac{\Lambda_3}{2\Lambda_1}}(\zeta + \vartheta)\right). \tag{2.3}$$

(ii) Trigonometric solutions: (at $\frac{\Lambda_3}{\Lambda_1} < 0$)

$$u_{5,6}(x, t) = \pm \sqrt{\frac{\Lambda_3}{\Lambda_2}} tan\left(\sqrt{\frac{-\Lambda_3}{2\Lambda_1}}(\zeta + \vartheta)\right) \tag{2.4}$$

and

$$u_{7,8}(x, t) = \pm \sqrt{\frac{\Lambda_3}{\Lambda_2}} cot\left(\sqrt{\frac{-\Lambda_3}{2\Lambda_1}}(\zeta + \vartheta)\right). \tag{2.5}$$

(iii) Rational solutions: (at $\Lambda_3 = 0$)

$$u_{9,10}(x, t) = \left(\mp \sqrt{\frac{-\Lambda_2}{2\Lambda_1}}(\zeta + \vartheta)\right)^{-1}. \tag{2.6}$$

Here $\vartheta$ is an arbitrary constant.

## 3 Solutions of MS

In this section, we introduce the stochastic solutions to MS (1.2) via Itô sense with multiplicative noise.

Using the transformation [29]:

$$U(x, y, t) = u(\zeta) e^{i(rx + \alpha y + \gamma t) + \sigma W(t) - \sigma^2 t}, \quad \Phi(x, y, t) = \Phi(\zeta), \; \zeta = x + \beta y - 2rt, \tag{3.1}$$

$r, \alpha, \gamma$ and $\beta$ are constants. Applying the same steps in [29], gives

$$\Phi(\zeta) = \left(\frac{1}{2r - \beta}\right) u^2(\zeta) \tag{3.2}$$

and

$$\Lambda_1 \, u''(\zeta) + \Lambda_2 \, u^3(\zeta) + \Lambda_3 \, u(\zeta) = 0, \tag{3.3}$$

where

$$\Lambda_1 = 1, \Lambda_2 = \frac{1}{2r - \beta}, \Lambda_3 = -(\gamma + r^2). \tag{3.4}$$

Thus the solutions of Eq (1.2) are:

$$u_{1,2}(\zeta) = \pm\sqrt{(\gamma + r^2)(2r - \beta)} \, tanh\left(\sqrt{\frac{-(\gamma + r^2)}{2}}\zeta\right), 2r - \beta < 0, \gamma + r^2 < 0. \tag{3.5}$$

Hence,

$$U_{1,2}(\zeta) = \pm\sqrt{(\gamma + r^2)(2r - \beta)} \, e^{i(rx + \alpha \, y + \gamma \, t) + \sigma \, W(t) - \sigma^2 \, t} \, tanh\left(\sqrt{\frac{-(\gamma + r^2)}{2}}(x + \beta y - 2rt)\right), \tag{3.6}$$

$2r - \beta < 0, \gamma + r^2 < 0.$

$$u_{3,4}(\zeta) = \pm\sqrt{(\gamma + r^2)(2r - \beta)} \, coth\left(\sqrt{\frac{-(\gamma + r^2)}{2}}\zeta\right), 2r - \beta < 0, \gamma + r^2 < 0. \tag{3.7}$$

Hence,

$$U_{3,4}(\zeta) = \pm\sqrt{(\gamma + r^2)(2r - \beta)} \, e^{i(rx + \alpha \, y + \gamma \, t) + \sigma \, W(t) - \sigma^2 \, t} \, coth\left(\sqrt{\frac{-(\gamma + r^2)}{2}}(x + \beta y - 2rt)\right), \tag{3.8}$$

$2r - \beta < 0, \gamma + r^2 < 0.$

$$u_{5,6}(\zeta) = \pm\sqrt{(\gamma + r^2)(\beta - 2r)} \, tan\left(\sqrt{\frac{(\gamma + r^2)}{2}}\zeta\right), \beta - 2r > 0, \gamma + r^2 > 0. \tag{3.9}$$

Hence,

$$U_{5,6}(x, y, t) = \pm\sqrt{(\gamma + r^2)(\beta - 2r)} \, e^{i(rx + \alpha \, y + \gamma \, t) + \sigma \, W(t) - \sigma^2 \, t} \, tan\left(\sqrt{\frac{(\gamma + r^2)}{2}}(x + \beta y - 2rt)\right), \tag{3.10}$$

$\beta - 2r > 0, \gamma + r^2 > 0.$

$$u_{7,8}(\zeta) = \pm\sqrt{(\gamma + r^2)(\beta - 2r)} \, cot\left(\sqrt{\frac{(\gamma + r^2)}{2}}\zeta\right), \beta - 2r > 0, \gamma + r^2 > 0. \tag{3.11}$$

Hence,

$$U_{7,8}(x, y, t) = \pm\sqrt{(\gamma + r^2)(\beta - 2r)} \, e^{i(rx + \alpha \, y + \gamma \, t) + \sigma \, W(t) - \sigma^2 \, t} \, cot\left(\sqrt{\frac{(\gamma + r^2)}{2}}(x + \beta y - 2rt)\right), \tag{3.12}$$

$\beta - 2r > 0, \gamma + r^2 > 0.$

$$u_{9,10}(\zeta) = \pm\sqrt{2(\beta - 2r)}\frac{1}{\vartheta + \zeta}, \qquad \beta - 2r > 0. \tag{3.13}$$

Hence,

$$U_{9,10}(x, y, t) = \pm\sqrt{2(\beta - 2r)}\frac{1}{\vartheta + x + \beta y - 2rt}\, e^{i(rx + \alpha\, y + \gamma\, t) + \sigma\, W(t) - \sigma^2\, t}, \; \beta - 2r > 0. \tag{3.14}$$

## 4 Physical interpretation

We have put into practice the cohesive method for identifying new significant stochastic forms in Itô sense for the coupled MS with multiplicative noise. This model is a sophisticated nonlinear model that is used in a variety of disciplines, including hydrodynamics, plasma physics, and nonlinear optics, to describe the dynamics of isolated waves that are contained in a tiny region of space [18, 32].

Most typical papers evaluated the suggested MS in deterministic situations. Unlike our approach, we investigate this model under the stochastic condition, i.e., in the Itô sense of being forced by multiplicative noise. A unified method was employed to discover some innovative random solutions to the MS through the Itô sense. This provided a range of dispersive and dissipative structures in the form solutions of Eq (3.3). The obtained solutions are hyperbolic, trigonometric and rational functions that explained a variety of remarkable physical phenomena in Bose-Einstein condensates, nonlinear optics, superconductivity, plasma and atmospheric physics, and other topics.

We provide some 2D and 3D graphs for some chosen solutions of the proposed model for appropriate parametric choices using Matlab release 18. In the absence of a noise term i.e., $\sigma = 0$, the solution (3.6) depicts periodic waves as in Figs 1 and 2. The solution (3.6) represents a random structural representation as illustrated in Figs 3 and 4. These figures illustrate how the dissipative solution (3.6) varies with space $x$, time $t$, and the noise effect $\sigma$. As $\sigma$ grows, the rate forcing wave rises and a high-amplitude shock random wave is produced, as seen in Fig 3.

Finally, the efficient, simple, succinct, and potent technique that is being used to extract accurate solitons may be extended to various nonlinear partial differential equations in mathematical physics, engineering, and applied sciences.

## 5 Conclusions

In this study, the unified technique is used acquiring some travelling wave solutions to the nonlinear Maccari's system in Itô sense with multiplicative noise. Significant new solitary waves are created. These solutions can prevent energy waves from collapsing or being forced. This applies to plasma physics, Langmuir solar wind, nonlinear optics and hydrodynamics. It was observed that increasing the random parameter induced both frenzied solitonic collapse and driving shock wave amplitude. To our knowledge, no prior material has been published that applies the unified technique to the model under consideration in this work. We'll employ different analytical techniques in subsequent work to obtain different kinds of solutions. Additionally, we may examine the Maccari model's bifurcation and chaotic patterns.

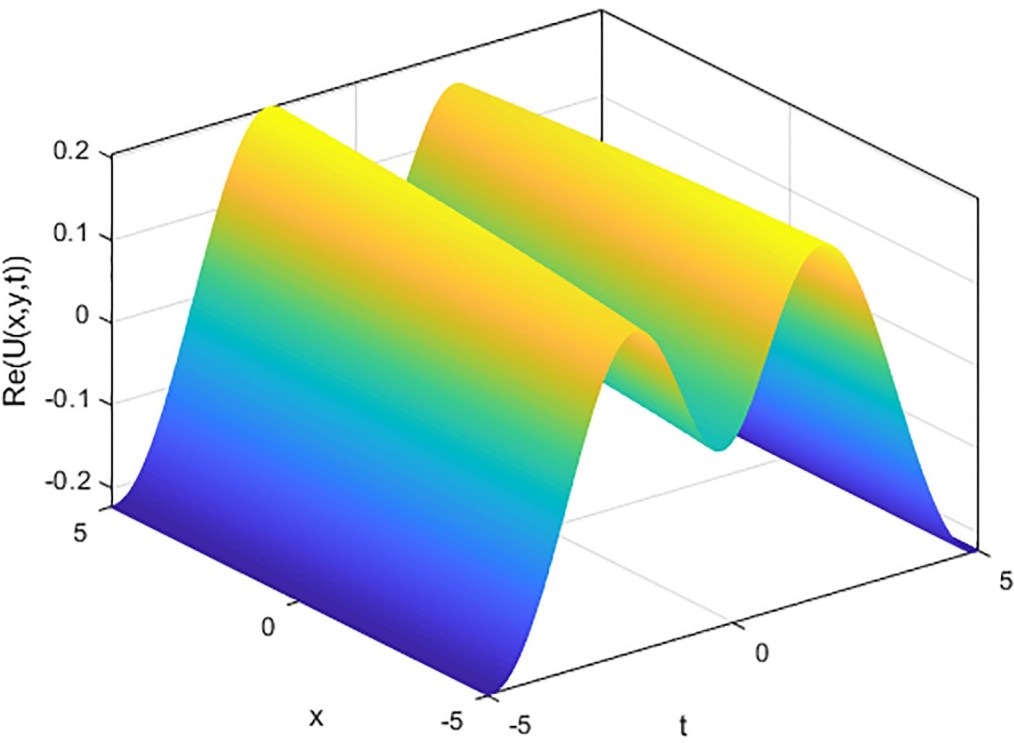

**Fig 1. 3D plot of soliton wave solution (3.6) for $\sigma = 0$.**

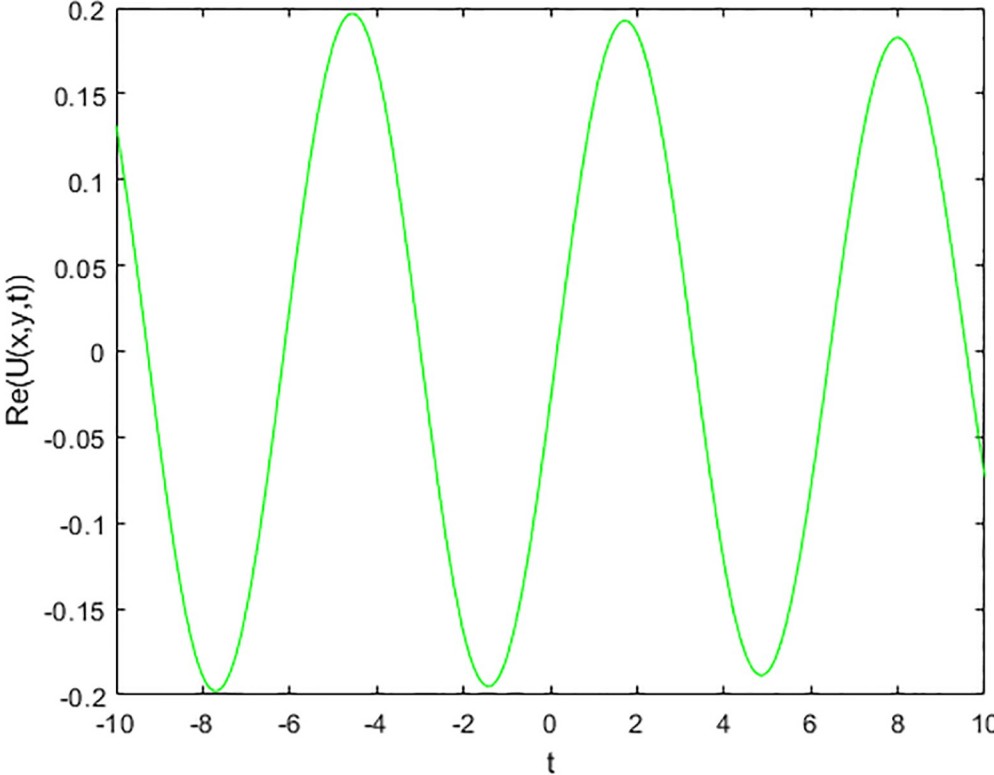

**Fig 2. 2D plot of soliton wave solution (3.6) for $\sigma = 0$.**

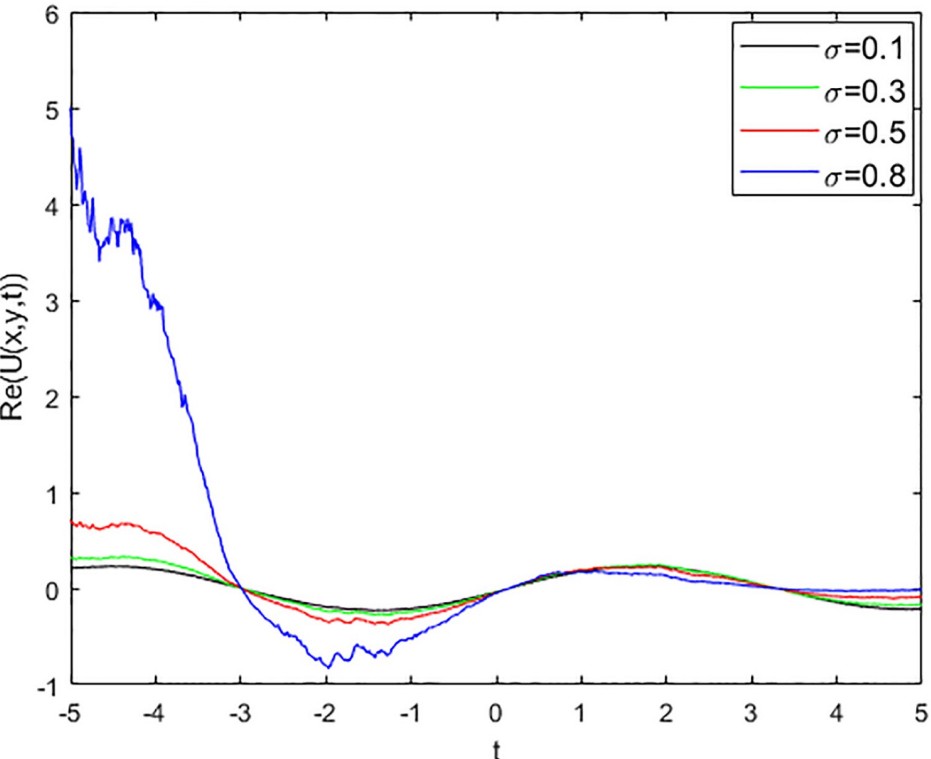

**Fig 3. 2D plot of soliton wave solution (3.6).**

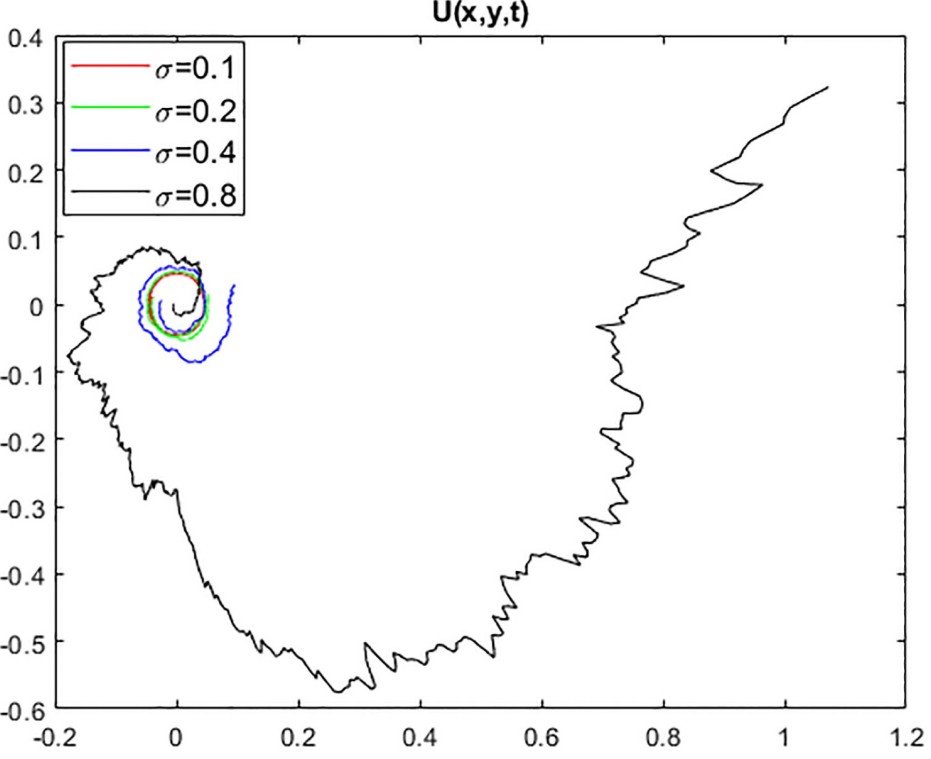

**Fig 4. Trajectory of solution (3.6) with noise strength $\sigma$.**

## Author Contributions

**Conceptualization:** Hesham G. Abdelwahed, Reem Alotaibi, Emad K. El-Shewy, Mahmoud A. E. Abdelrahman.

**Data curation:** Reem Alotaibi.

**Formal analysis:** Hesham G. Abdelwahed, Emad K. El-Shewy, Mahmoud A. E. Abdelrahman.

**Software:** Hesham G. Abdelwahed, Emad K. El-Shewy, Mahmoud A. E. Abdelrahman.

**Writing – original draft:** Hesham G. Abdelwahed, Reem Alotaibi, Emad K. El-Shewy.

**Writing – review & editing:** Mahmoud A. E. Abdelrahman.

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
