## [Decision Letter · Decision Letter 0]

31 Jul 2024

PONE-D-24-28903The structure of stochastic solutions for Maccarimodel via unified techniquePLOS ONE

Dear Dr. Gomaa,

Thank you for submitting your manuscript to PLOS ONE. After careful consideration, we feel that it has merit but does not fully meet PLOS ONE’s publication criteria as it currently stands. Therefore, we invite you to submit a revised version of the manuscript that addresses the points raised during the review process.

We look forward to receiving your revised manuscript.

Kind regards,

Rajesh Sharma

Academic Editor

PLOS ONE

Journal Requirements:

3. We note that your Data Availability Statement is currently as follows: 

"All relevant data are within the manuscript and its Supporting Information files."

**Additional Editor Comments:**

We note that one or more reviewers has recommended that you cite specific previously published works. As always, we recommend that you please review and evaluate the requested works to determine whether they are relevant and should be cited. It is not a requirement to cite these works. We appreciate your attention to this request.

Reviewers' comments:

Reviewer's Responses to Questions

**Comments to the Author**

1. Is the manuscript technically sound, and do the data support the conclusions?

Reviewer #1: Yes

Reviewer #2: Yes

Reviewer #3: Yes

2. Has the statistical analysis been performed appropriately and rigorously? 

Reviewer #1: Yes

Reviewer #2: N/A

Reviewer #3: Yes

3. Have the authors made all data underlying the findings in their manuscript fully available?

Reviewer #1: Yes

Reviewer #2: Yes

Reviewer #3: Yes

4. Is the manuscript presented in an intelligible fashion and written in standard English?

Reviewer #1: Yes

Reviewer #2: Yes

Reviewer #3: Yes

5. Review Comments to the Author

Reviewer #1: Reviewer Comments:

Dear Authors / Editor

They investigates the unified approach is used in acquiring some new results to the coupled Maccari system (MS) in Itˆo sense with multiplicative noise. The MS is a nonlinear model used in hydrodynamics, plasma physics, and nonlinear op tics to represent isolated waves in a restricted region. We provide new results with complicated structures to this model, including hyperbolic, trigonometric and ra tional function solutions. We draw the two dimensional (2D) and three dimensional (3D) graphs to some of the study’s solutions under appropriately chosen physical parameter values. Random factors can alter the collapse caused by turbulence in the model medium. We noticed that our results may be useful for solving some real-world physical issues by identifying the motion of an isolated wave in a small area. Some points should be discussed and improved:

1) A comparative study should be included. What is your innovative work? Did you compare your results with others authors results?

2) The introduction should write according to the following questions: What is already known in the open literature? What is missing (i.e., research gaps)? What needs to be done, why, and how? Clear statements of the novelty of the work should also appear briefly in the Abstract and Conclusions sections. The authors should amend or revise their manuscript based on the following literature. You should cite your paper 2020 to 2023 recent papers: For better presentation of the paper, the authors should cite and discuss recent works on the solitons in this field:

1. Md Mamunur Roshid, Md. Nur Alam, Onur Alp İlhan, Md. Abdur Rahim, Md. Mehedi Hassen Tuhin, M.M. Rahman, Modulation Instability and Comparative Observation of The Effect of Fractional Parameters on New Optical Soliton Solutions of The Paraxial Wave Model, Optical and Quantum Electronics, 56, 1010 (2024). DOI: https://doi.org/10.1007/s11082-024-06921-7

2. Mujahid Iqbal, Md. Nur Alam, Dianchen Lu, Aly R. Seadawy, Nahaa E. Alsubaie, and Salisu Ibrahim, Applications of nonlinear longitudinal wave equation with periodic optical solitons wave structure in magneto electro elastic circular rod, Optical and Quantum Electronics (2024) 56:1031 DOI: https://doi.org/10.1007/s11082-024-06671-6

We recommend that you revise your manuscript according to these comments and resubmit it for further consideration. We appreciate your time and effort, and we look forward to receiving your revised manuscript.

Reviewer #2: Dear authors,

The report of your paper entitled "The structure of stochastic solutions for Maccari model via unified technique" is attached to this submission. Some comments need to be considered for your paper.

Best wishes

Reviewer

Reviewer #3: In this paper, the author considered the unified approach to produce some new results to the coupled ‎Maccari system (MS) in Ito sense with multiplicative noise. They provide new results with complicated ‎structures to this model, including hyperbolic, trigonometric, and rational function solutions. She ‎draws the 2D and 3D graphs to some of the study’s solutions under appropriately chosen parameter ‎values. Random factors can alter the collapse caused by turbulence in the model medium. After ‎reviewing the article, I believe it is technically correct, insightful, and suitable for publishing in this ‎respectable journal. However, I propose that certain shortcomings or flaws be resolved adequately.‎

The structure of stochastic solutions for Maccari model via unified technique

‎1- The title of this research manuscript should read “An insight into the stochastic solitonic features of the ‎Maccari model using the solver technique” instead of The structure of stochastic solutions for Maccari ‎model via unified technique”. ‎

‎2- What are the key features of the proposed technique? ‎

‎3-The significance of the obtained solutions should be highlighted. ‎

‎4- In the introduction section, authors should present some properties of Browning motion process.‎

‎5-The authors must clearly indicate what is the difference between the present work history and other ‎research work.‎

‎6- Revise the whole manuscript for punctuation issues, such as end of equations (3.2).‎

‎7- The authors should state in the paper what kind of software package has been used to obtain the required ‎results. ‎

‎8- The entire manuscript should be totally double-checked for typographical errors, text inconsistencies.‎

In short, the results of the paper are satisfactory. After correction along the above lines, the paper can be ‎accepted for publication.‎

6. PLOS authors have the option to publish the peer review history of their article (what does this mean?). If published, this will include your full peer review and any attached files.

Reviewer #1: No

Reviewer #2: No

Reviewer #3: No

---

## [Author Response · Author response to Decision Letter 0]

2 Sep 2024

Kindly find the attached file ''Detailed Responses to Reviewers''

---

## [Decision Letter · Decision Letter 1]

14 Oct 2024

An insight into the stochastic solitonic features of theMaccari model using the solver technique

PONE-D-24-28903R1

Dear Dr. Gomaa,

We’re pleased to inform you that your manuscript has been judged scientifically suitable for publication and will be formally accepted for publication once it meets all outstanding technical requirements.

Kind regards,

Rajesh Sharma

Academic Editor

PLOS ONE

Additional Editor Comments (optional):

Reviewers' comments:

Reviewer's Responses to Questions

**Comments to the Author**

1. If the authors have adequately addressed your comments raised in a previous round of review and you feel that this manuscript is now acceptable for publication, you may indicate that here to bypass the “Comments to the Author” section, enter your conflict of interest statement in the “Confidential to Editor” section, and submit your "Accept" recommendation.

Reviewer #2: All comments have been addressed

Reviewer #3: All comments have been addressed

2. Is the manuscript technically sound, and do the data support the conclusions?

Reviewer #2: Partly

Reviewer #3: Yes

3. Has the statistical analysis been performed appropriately and rigorously? 

Reviewer #2: N/A

Reviewer #3: Yes

4. Have the authors made all data underlying the findings in their manuscript fully available?

Reviewer #2: Yes

Reviewer #3: Yes

5. Is the manuscript presented in an intelligible fashion and written in standard English?

Reviewer #2: Yes

Reviewer #3: Yes

6. Review Comments to the Author

Reviewer #2: Dear authors,

The provided comments have been successfully achieved. The paper is now fine.

Best wishes

Reviewer #3: it is now Ok. i suggest to be accepted

7. PLOS authors have the option to publish the peer review history of their article (what does this mean?). If published, this will include your full peer review and any attached files.

Reviewer #2: No

Reviewer #3: **Yes: **Hossam A. Nabwey

---

## [Editor Report · Acceptance letter]

25 Oct 2024

PONE-D-24-28903R1 

PLOS ONE

Dear Dr. Abdelwahed, 

I'm pleased to inform you that your manuscript has been deemed suitable for publication in PLOS ONE. Congratulations! Your manuscript is now being handed over to our production team.

Kind regards, 

on behalf of

Dr. Rajesh Sharma 

Academic Editor

PLOS ONE